# The Pandemic-Related Factors Associated with Emergency Department Visits in Portugal throughout Two Years of the Pandemic: A Retrospective Population-Based Study

**DOI:** 10.3390/ijerph20021207

**Published:** 2023-01-10

**Authors:** Walaa Kinaan, Patrícia Soares, João Victor Rocha, Paulo Boto, Rui Santana, Sílvia Lopes

**Affiliations:** 1NOVA National School of Public Health, NOVA University Lisbon, 1600-560 Lisboa, Portugal; 2NOVA National School of Public Health, Public Health Research Centre, Comprehensive Health Research Center (CHRC), NOVA University Lisbon, 1600-560 Lisboa, Portugal

**Keywords:** COVID-19 pandemic, emergency department, incidence, vaccination, Portugal

## Abstract

The COVID-19 pandemic has affected the use of emergency departments (ED) worldwide. This study identifies the pandemic-related factors associated with the number of ED visits in mainland Portugal and each of its regions. We collected data on ED visits from March 2020 to March 2022. Data on incidence, vaccination, mobility, containment index, and Google search volume were retrieved from open online sources at different time points. We fitted a quasi-Poisson generalized linear regression model, and each variable was modeled separately and adjusted for time and month. There was a positive ED trend throughout the two years of the pandemic in mainland Portugal and each of its regions. In the mainland, during months with high workplace mobility, there were 10.5% more ED visits compared to months with average mobility. ED visits decreased in months with low mobility for retail and recreation, groceries and pharmacies, and transit compared to months of medium mobility. Portugal saw a reduction in ED utilization during the pandemic period, but with a positive trend from March 2020 to March 2022. The change in the population’s behavior of seeking the ED throughout the pandemic might be associated with mobility, incidence, and pandemic fatigue.

## 1. Introduction

Since the beginning of the COVID-19 pandemic, concerns have arisen about changes in the utilization of health services [1,2,3], such as the reduced use of emergency departments (ED), with reduction rates varying from 26.8 to 63.8% early in the pandemic [4,5,6]. This phenomenon was also documented in different countries in Europe, specifically in Spain [7], Italy [8], and Germany [4]. Similarly, in Portugal, reductions of 57% and 48% were reported for all hospitalizations and emergency visits, respectively [9,10]. The first study predicted inpatient hospital admissions for March to May 2020, based on historical records, and compared this prediction with the observed number of hospitalizations and its characteristics, while the second study addressed the use of emergency services during the first pandemic month, compared to historical records. Both studies highlighted these reductions in the early periods of the pandemic. However, neither of these studies sought to identify factors associated with emergency department visits during the pandemic in order to describe the pandemic effect. A study from the Netherlands reported that ED volume reduced differently during the pandemic period, where ED visits decreased by 18% during the study period in comparison with the same period in 2019 and further declined (−29%) during the lockdown [11].

The striking decline in ED visits likely resulted from various factors, such as government policies during the pandemic and population compliance with preventive isolation measures. The indirect effect of the rigorous mitigation strategies imposed to flatten the epidemic curve may have unintentionally dissuaded patients from visiting EDs [11]. In many European countries, it has been reported that stay-at-home orders, lockdowns, and restricted access to public spaces may have discouraged the public from using ED services [7,11]. The population’s compliance with these measures might also have resulted in falling mobility trends, as reported by Google [12], and a reduction in traffic and workplace accidents [7].

Beyond containment measures, some authors attempted to explain this reduction by the changes in health-seeking behavior. The fear of acquiring the virus in the hospitals [7,11], perhaps due to the high incidence rates, reluctance to seek healthcare [13], a lack of understanding of the transmission of the virus [14], the shift to telehealth [14], changes in healthcare provider behavior [15], and possibly low vaccination coverage are factors that could motivate changes in behavior during the crisis and may have contributed to changes in people’s willingness to use the ED.

We hypothesize that in the Portuguese context, containment measures and changes in population behavior, among many other factors, could have induced an attitude favoring reduced ED visits. This is likely because, as in many European countries, the Portuguese government had imposed stringent measures to contain the effects of the pandemic [16]. The Portuguese population not only showed signs of adhering to the measures by reducing mobility trends, as reported by Google [12], but also exhibited a fear of visiting health services, delaying their healthcare [17], which could indicate changes in their health-seeking behaviors. Thus, we aimed to identify the pandemic-related factors associated with the number of ED visits in mainland Portugal and each region.

## 2. Materials and Methods

### 2.1. Study Design and Study Period

We conducted a retrospective study on the association between the characteristics of the pandemic period (COVID-19 incidence and vaccination, population mobility, containment index as an indicator of restrictions imposed by the government, and Google search volume to assess public awareness of COVID-19) and the frequency of ED visits in public hospitals, with months used as a unit of analysis. We studied mainland Portugal and its five regions during the pandemic period, from March 2020 to March 2022. The first COVID-19 case in Portugal was confirmed on 2 March 2020, while March 2022 was the most recent month for which data regarding our variable of interest (frequency of ED visits) were available at the time of extraction.

### 2.2. Data Sources

The number of ED visits was extracted from the Portuguese National Health Service Transparency Portal on 11 May 2022 [18]. This is an open online source with data on the cumulative monthly frequency of ED visits per hospital and region in mainland Portugal. All ED visits within the defined time frame were included.

Data on daily cases of COVID-19 and weekly vaccination rates for the mainland and the regions were extracted from the GitHub repository on 30 June 2022, and 21 March 2022, respectively [19]. This repository extracts data from the Portuguese Directorate General of Health. The vaccination data were available until November 2021; when reporting ceased, the coverage was above 86% in the mainland [19].

The estimates of the resident population in mainland Portugal and each region were downloaded from Statistics Portugal on 11 May 2022 [20]. Since population estimates for 2021 and 2022 were unavailable, we extracted data for the most recent year available (2020).

The Oxford Coronavirus Government Response Tracker (OxCGRT) [21] was consulted on 13 May 2022 to extract information on government restrictions applicable to Portugal on each day of the study period. This index ranges from 0 to 100 based on the government lockdown intensity. Higher values represent stricter government policies.

We extracted mobility trends per day for Portugal from the COVID-19 Google mobility trends report [12]. This dataset provides anonymized data obtained from Google services such as Google maps. It represents the percentage change in population movement at each specific location compared to that on baseline days before the COVID-19 pandemic (3 January–6 February 2020) in which a positive value of mobility changes indicates an increase in movement at the corresponding location category.

On 29 April 2022, we extracted the weekly Google search volume on the term “coronavirus” in Portugal, as a proxy for COVID-19 awareness [22]; on this scale, the numbers represent the search interest relative to a specific topic. Google trends analyzes an unbiased sample of Google web searches and divides it by the total number of the queries made at a certain time in a specific location. Subsequently, the resulting numbers are scaled on a range from 0 to 100, with higher values indicating greater interest.

### 2.3. Measurements

#### 2.3.1. Outcomes

As outcome variables, we considered the number of ED visits on the mainland and in each region (North, Centre, Lisbon and Tagus Valley, Alentejo, and Algarve). All these outcomes were measured by the total number of monthly ED visits, computed from monthly cumulative values.

#### 2.3.2. Independent Variables

The monthly incidence rate of COVID-19 per 10,000 inhabitants and the proportion of the population vaccinated against COVID-19 were computed. Data on the daily containment index were reported as the maximum value for each month in the study period.

For mobility, we studied Google mobility trends for retail and recreation locations (RR) (restaurants, cafes, shopping centres, theme parks, museums, libraries, and movie theatres), grocery stores and pharmacies (GP) (grocery markets, food warehouses, farmers markets, speciality food shops, drugstores, and pharmacies), transit stations (public transport hubs such as subway, bus, and train stations), and workplaces. There was no information on these variables for the regions; hence, they were analysed only for Portugal. The maximum value was recorded for each corresponding month. The same rationale was applied to Google search volume data, which were available only for Portugal. We used the maximum values to describe the worst moment during the month.

All independent variables were categorized to facilitate interpretation. Except for vaccination, the remaining variables were classified into terciles (low, medium, and high). Since vaccination followed specific phases during the pandemic, we coded it into three categories (“no vaccination,” “start of vaccination,” and “stabilization”). The cutoff for the stabilization of vaccination was estimated for the mainland and each region, based on the percentage change in the monthly COVID-19 vaccination rate. Stabilization was assumed when two consecutive months showed a decrease in the percentage change.

### 2.4. Statistical Analysis

We reported summary statistics of the study variables as the median and the percentiles (25 and 75) or the median and the range (maximum, minimum) in each category. We also calculated the median number of ED visits and interquartile range (IQR) for each variable of interest. Our study occurred over two years, between March 2020 and March 2022, corresponding to 24 time points. Due to the overdispersion of the outcomes, we fitted a quasi-Poisson generalized linear regression model. Each variable of interest was modeled separately and adjusted for time, corresponding to the trend during the pandemic, and month as a spline variable. The reference groups for the independent variables were chosen to represent the scenario closest to normality (low incidence, vaccination stabilized, low index of restrictions, low Google search volume, and medium mobility index). We reported the effect of these variables as incidence rate ratios (IRR), with a 95% confidence interval (95% CI). We discussed the results as a percentage increase or decrease to facilitate understanding. Diagnostics, plotting residuals, autocorrelation, and partial autocorrelation functions were performed for each regression model. All analyses were performed using R 4.2.1 [23].

## 3. Results

### 3.1. Study Variables

The characteristics of the variables are presented for the mainland (Table 1) and the regions (Table 2). Half of the months recorded at least 380,169 ED visits in the mainland (IQR: 328,884–464,062) (Appendix A), where the median incidence of COVID-19 per 10,000 inhabitants ranged from 10 in the low incidence months to 264 in the high incidence months (Table 1) (Appendix A). As for the vaccination rate, at least 84% of the eligible population was vaccinated during the stabilization period (Appendix A). The containment index reached a maximum of 85, and the median ranged from 58 to 78 in each tercile (Appendix A). Mobility for retail and recreation was above pre-pandemic values only in the category “High RR mobility trend” (−29, −6, and 6). That was not the case for grocery and pharmacy mobility, which was below pre-pandemic values only in the “Low GP mobility trend” category (−5, 15, and 38). Transit mobility presented the largest difference compared to pre-pandemic mobility and did not recover to pre-pandemic values (−43, −24, −5). The same also occurred for workplace mobility (−24, −14, −6) (Appendix A). The median Google searches on the term ‘Coronavirus’ ranged from 16 in the low search volume to 42 in the high search volume (Appendix A).

In the analysis per region, we observed a considerable difference between incidence in low and high incidence months. The median in the lowest tercile was between 5 (Centre and Alentejo) and 14 (Lisbon and Tagus Valley) cases per 10,000 inhabitants and between 218 (Alentejo) and 329 (Lisbon and Tagus Valley) in the highest tercile (Table 2) (Appendix A). The median of ED visits ranged between 21,814 in Alentejo and 135,435 in the North (Appendix A). The stabilization of vaccination started as low as 77.7% for Alentejo and as high as 87.9% for the North (Appendix A).

### 3.2. The Pandemic Factors Associated with ED Visits in the Mainland

The months for which we observed the lowest frequency of ED were those with low incidence, no vaccination, high containment, low mobility, and medium Google search volumes (Table 3). ED admissions in the mainland increased significantly (by 2.6%) during the study period (IRR: 1.02, 95% CI: 1.02–1.03). ED visits fell significantly in the months with low mobility for retail and recreation (IRR: 0.876, 95% CI: 0.80–0.96), groceries and pharmacies (IRR: 0.88, 95% CI: 0.79–0.98), and transit (IRR: 0.87, 95% CI: 0.79–0.97) when compared to months of medium mobility. During months with high workplace mobility, there were 10.5% more ED visits compared to months with medium mobility (IRR: 1.10, 95% CI: 1.00–1.21). The remaining variables (incidence, vaccination, containment, and Google search volume) were not significantly associated with the number of ED visits.

### 3.3. Characteristics Associated with ED Visits by Region

ED visits were the lowest in all regions in months with low COVID-19 incidence. Similarly, they were the lowest during the months with no vaccination against COVID-19 (Table 4). We observed a trend of growing ED demand similar to that observed in the mainland, between 2.5% (95% CI: 1.02–1.03) and 3.0% (95% CI: 1.02–1.03) in the North and Algarve regions, respectively. However, the only significant variation was found in the North region, where ED visits fell significantly during medium incidence months compared to the months in the low COVID-19 incidence category (IRR: 0.88, 95% CI: 0.80–0.98). There were no significant variations in ED demand between different levels of vaccination.

## 4. Discussion

Although earlier studies report a reduction in ED demand during the pandemic period [5,24,25], the factors associated with this reduction have not been well addressed in the literature. Herein, we have sought to identify the pandemic-related factors associated with the number of ED visits in mainland Portugal and each of its regions during two years in the COVID-19 era. We found that: (I) ED visits increased in the mainland and the regions during the study period, (II) there was a decrease in ED visits in months with low mobility in retail and recreation locations, transit stations, and groceries and pharmacies, (III) there was an increase in ED visits in months with high mobility in workplaces, and (IV) ED visits in the North region were lower during the months with medium incidence rates than in the months with low incidence rates.

To the best of our knowledge, this is the first study to explore a positive trend in ED visits lasting over a relatively long period of time (two years) of the pandemic and showing that population behaviors constantly evolved. Recent findings from 14 countries suggest a “temporal difference in adherence to protective behaviors against COVID-19” [26]. This phenomenon might also be present in the utilization of health services. The tendency to return to “normal life,” pandemic fatigue, and community vaccination, among other factors, could have contributed (to some extent) to this trend. Although our results were not significant, ED visits were more numerous during the months of vaccination stabilization than before vaccination started, which could mean that vaccination increased the population’s confidence in returning to normal life. Another possible explanation is pandemic fatigue, which can be defined as a natural reaction to a long-lasting pandemic [27], in which the population may have experienced boredom or fatigue with social restrictions and other risk mitigation strategies [28]. In that case, the increase in ED utilization might not reflect an easing of fear of using the health services, but rather the fatigue from the prolonged containment measures and the comfort of high vaccination rates. However, more studies are needed to confirm or refute these speculations.

In our study, we observed a decrease in mobility in retail and recreation locations, groceries and pharmacies, and transit stations, as well as a similar falling trend in ED demand. Similarly, an upward trend in ED visits was associated with increased workplace mobility. This is in line with a study conducted in Finland [29], in which increased ED visits were associated with a return to normalcy in traffic activities. The reduced mobility may have minimized the probability of injuries, resulting in reductions in the incidence of events such as traffic and car accidents [30,31], fewer traumatic injuries [32], and a lower volume of work-related injuries due to the widespread adoption of remote work [33]. Reduced nightlife-related ED visits, such as intoxications [11], and reduced face-to-face contacts [34] might also have played a substantial role in decreased ED visits.

Our study has shown a decrease in ED visits during the months with medium incidence rates in the North compared to low incidence months. In Portugal, the reduced demand for emergency procedures during the pandemic was correlated with increased COVID-19 cases [35]. Likewise, in Taiwan, ED visits were reduced during the pandemic period and returned to an increased level during the period with “no change in community infection” [36]. The reluctance to use the ED during moderate-incident months may reflect the fear of infection [33]. Uncertainty in pandemic times has perhaps discouraged the public from using ED because they likely desired to reduce the burden on the healthcare system [37]. However, our findings suggest that such associations were not present for the other regions, perhaps because this effect was undetected by the model or due to regional variations in the stringency of policies implemented in 2021 (unlike in 2020, when policies were imposed nationwide). However, this is difficult to assess empirically, and further research is needed.

A previous study has shown a significant association between ED visits and the stringency of social distancing policies [38], but our study did not find this association. One could hypothesize that perhaps public health policies reduced access to health services and provoked fear of using such services [29]. As a result, patients may have sought private healthcare units or health centers outside the hospital ED.

In contrast to a previous study [39], ED visits were more numerous during the months with increased Google search volumes; however, these results were not significant. The extent to which Google search volumes can be used to predict health-seeking behavior is still vague. One might surmise that perhaps the Google search in our study indicates COVID-19 public awareness motivated by media coverage, as it has been previously reported [39] or curiosity about the disease, rather than fear or concerns. Similarly, the volume of searches could also reflect the population’s desire to become informed about COVID-19 before accessing emergency services.

This study has several strengths. To the best of our knowledge, this is one of the first studies in Portugal—or any other country—to examine the factors associated with ED utilization patterns during the pandemic over a relatively long period (two years). Our study demonstrates the feasibility of using large web-based services, such as Google search, and mobility to capture available information and employ it to predict and recognize behavior-based patterns in the population. Nevertheless, these data are imperfect proxies for accurate behavioral patterns and are not representative of the general population. The study’s limitations include the inclusion of few data points, and a lack of data regarding the containment index and the Google search and mobility for the mainland or the regions. Google mobility data is limited to those accessing the Google map application and allowing Google to record their location history [40]. Although the dataset provides a comprehensive overview of ED visits, the ED visits in our study do not include all of the ED visits in mainland Portugal. In 2020, ED visits in public hospitals accounted for 77.4% of the total ED visits [41]. Moreover, using the maximum values for analysis might be an overestimation, affecting our results. Our study also lacks data on the proportion of Google maps users, hence the representativeness of our study is undefined. Finally, because it is an ecological study, it is not possible to infer causality between the variables and ED visits. Nevertheless, our study indicates that mobility and incidence were associated with reductions in ED visits during the two years of the pandemic.

It is likely that some patients deferred using the ED services and delayed necessary medical care due to fear of COVID-19, thereby resulting in increased morbidity and mortality [42]. Therefore, understanding the factors associated with reduced ED visits during a pandemic might help tailor the implementation of measures to promote public health in current and future pandemics. Our study highlights the importance of future extension of this work on the factors associated with changes in health-seeking behaviors regarding ED utilization, including pandemic fatigue. It also showed that the non-health needs-related factors are of significant importance for ED demand in the pandemic context. The upward ED trend shows that the early phase of the pandemic is the most critical period, and health authorities should pay special attention to ensure that health needs are met during this period particularly. Additionally, the regional differences in the association between pandemic factors and ED demand call for appropriate measures and/or research adapted to each local context. Finally, the challenges concerning finding an association between pandemic factors and ED reduction, including the limitations of our variables, highlights the fact that ED demand is a multifactorial and complex phenomenon.

## 5. Conclusions

Portugal saw a positive trend in ED utilization from March 2020 to March 2022. This article offers critical insights into the change in population behaviors of seeking the ED throughout the pandemic in Portugal and the opportunity to design healthcare interventions. The factors associated with the avoidance of the ED use included reduced mobility, a medium incidence rate of COVID-19, and strict government regulations, among others.

## Figures and Tables

**Table 1 ijerph-20-01207-t001:** Summary of the variables at national level.

Variable	Median (Min, Max)
Number of ED visits *	380,169 (328,884, 464,062)
COVID-19 incidence per 10,000 inhabitants	
Low incidence	10 (7, 17)
Medium incidence	43 (18, 83)
High incidence	264 (89, 1211)
COVID-19 vaccination	
No vaccination	0%
Start of vaccination	14% (0.3%, 73.8%)
Stabilization of vaccination	≥84%
Containment index	
Low containment	58 (43, 59)
Medium containment	70 (65, 73)
High containment	78 (74, 85)
Google searches ‘Coronavirus’ in Portugal	
Low search volume	16 (7, 20)
Medium search volume	24 (21, 31)
High search volume	42 (32, 100)
Mobility—retail and recreation (RR)	
Low RR mobility	−29 (−67, −15)
Medium RR mobility	−6 (−14, 2)
High RR mobility	6 (4, 18)
Mobility—grocery and pharmacy (GP)	
Low GP mobility	−5 (−35, 2)
Medium GP mobility	15 (3, 34)
High GP mobility	38 (34, 61)
Mobility—transit	
Low transit mobility	−43 (−71, −35)
Medium transit mobility	−24 (−33, −17)
High transit mobility	−5 (−9, 4)
Mobility—workplace	
Low workplace mobility	−24 (−55, −19)
Medium workplace mobility	−14 (−17, −10)
High workplace mobility	−6 (−8, 2)

* Median (IQR).

**Table 2 ijerph-20-01207-t002:** Summary of the variables in each region.

Variable	Median (Min, Max)				
	North	Centre	Lisbon and Tagus Valley	Alentejo	Algarve
Number of ED visits *	135,435 (114,516, 159,477)	70,680 (60,074, 82,399)	134,808 (117,052, 167,320)	21,814 (18,796, 26,584)	23,129 (18,241, 27,473)
COVID-19 incidence per 10,000 inhabitants					
Low incidence	12 (2, 15)	5 (2, 12)	14 (5, 22)	5 (1, 13)	6 (1, 14)
Medium incidence	37 (17, 83)	29 (12, 67)	55 (23, 102)	31 (15, 74)	52 (23, 69)
High incidence	235 (102, 1436)	245 (68, 925)	329 (111, 1216)	218 (76, 829)	229 (103, 966)
COVID-19 vaccination					
No vaccination	0%	0%	0%	0%	0%
Start of vaccination	19.8% (0.4%, 86.2%)	17.4% (0.4%, 75.2%)	12.4% (0.3%, 72.7%)	12.6% (0.2%, 67.2%)	18.8% (0.2%, 74.2%)
Stabilization of vaccination	≥87.9%	≥85.4%	≥81.8%	≥77.7%	≥85.3%

* Median (IQR).

**Table 3 ijerph-20-01207-t003:** ED visits for the variables and regression analysis for ED visits during the pandemic in mainland Portugal.

Variable	Median (IQR)	IRR ^1^ (95% CI ^1^)
Trend	—	1.02 (1.021, 1.03) ***
COVID-19 incidence per 10,000 inhabitants		
Low incidence	328,884 (322,594, 375,439)	—
Medium incidence	466,508 (380,068, 480,972)	0.914 (0.78, 1.06)
High incidence	420,541 (332,923, 468,516)	0.99 (0.84, 1.14)
COVID-19 vaccination		
Stabilization vaccination (78–88%)	464,062 (447,265, 506,753)	—
No vaccination	332,314 (324,438, 378,309)	1.09 (0.82, 1.46)
Start of vaccination	412,204 (321,052, 471,070)	0.99 (0.85, 1.15)
Containment index		
Low containment	409,603 (380,169, 491,642)	—
Medium containment	356,649 (329,062, 463,304)	0.94 (0.85, 1.04)
High containment	349,016 (314,160, 453,966)	0.94 (0.85, 1.04)
Google searches ‘Coronavirus’ in Portugal		
Low search volume	398,228 (374,311, 460,289)	—
Medium search volume	333,531 (319,227, 428,792)	0.94 (0.84, 1.04)
High search volume	405,623 (321,324, 464,527)	1.02 (0.93, 1.12)
Mobility—retail and recreation (RR)		
Medium mobility RR	383,510 (363,834, 435,852)	—
Low mobility RR	322,594 (307,356, 331,098)	0.87 (0.80, 0.96) **
High mobility RR	481,164 (463,809, 499,198)	0.97 (0.86, 1.08)
Mobility—grocery and pharmacy (GP)		
Medium mobility GP	383,310 (364,962, 453,966)	—
Low mobility GP	322,594 (307,356, 331,098)	0.88 (0.79, 0.98) *
High mobility GP	470,739 (455,158, 486,594)	0.99 (0.88, 1.10)
Mobility—transit		
Medium mobility retail	409,166 (380,068, 453,966)	—
Low mobility retail	322,594 (307,356, 331,098)	0.87 (0.79, 0.97) *
High mobility retail	470,739 (449,689, 499,198)	1.01 (0.92, 1.10)
Mobility—workplace		
Medium mobility workplace	379,968 (364,962, 439,625)	—
Low mobility workplace	322,594 (307,356, 373,935)	0.94 (0.85, 1.03)
High mobility workplace	466,003 (439,128, 499,198)	1.10 (1.00, 1.21) *

^1^ IRR = incidence rate ratio; CI = confidence interval; *** *p* < 0.001; ** *p* < 0.001; * *p* < 0.05.

**Table 4 ijerph-20-01207-t004:** ED visits for the variables and regression analysis for ED visits during the pandemic in the five regions.

Variable	Median (IQR)					IRR ^1^ (95% CI ^1^)				
	North	Centre	Lisbon and Tagus Valley	Alentejo	Algarve	North	Centre	Lisbon and Tagus Valley	Alentejo	Algarve
Trend						1.02 (1.02, 1.03) *	1.02 (1.02, 1.03) *	1.02 (1.02, 1.03) *	1.02 (1.020, 1.033) *	1.03 (1.023, 1.037) *
COVID-19 incidence per 10,000 inhabitants										
Low incidence	130,547 (114,512, 135,435)	68,375 (60,070, 70,680)	126,768 (111,489, 134,106)	19,681 (18,614, 21,814)	18,614 (17,909, 23,129)					
Medium incidence	163,074 (119,283, 168,426)	83,638 (62,259, 88,625)	148,145 (131,202, 168,147)	26,008 (21,106, 27,688)	20,400 (18,072, 27,888)	0.88 (0.80, 0.98) **	0.90 (0.81, 1.00)	0.96 (0.84, 1.08)	0.90 (0.77, 1.04)	0.92 (0.79, 1.06)
High incidence	139,722 (117,244, 154,296)	76,440 (60,973, 83,007)	145,310 (115,597, 169,744)	25,744 (18,968, 27,085)	26,284 (24,820, 30,470)	0.99 (0.88, 1.11)	1.04 (0.88, 1.24)	0.94 (0.81, 1.08)	0.92 (0.76, 1.11)	1.03 (0.85, 1.25)
COVID-19 vaccination										
Stabilization (78–88%) of vaccination	163,568 (154,296, 174,914)	82,399 (78,885, 90,998)	171,568 (158,974, 184,940)	26,766 (26,350, 29,100)	27,473 (24,966, 29,562)					
No vaccination	116,706 (109,005, 132,928)	60,688 (58,468, 68,302)	119,974 (116,627, 129,063)	19,396 (18,400, 20,821)	18,430 (17,724, 21,757)	0.99 (0.76, 1.30)	1.18 (0.88, 1.58)	0.99 (0.71, 1.38)	1.10 (0.79, 1.53)	1.20 (0.85, 1.71)
Start of vaccination	152,357 (114,512, 159,949)	74,739 (59,585, 85,806)	147,120 (111,425, 166,347)	23,270 (18,596, 26,308)	22,982 (17,887, 28,388)	0.93 (0.81, 1.06)	1.02 (0.88, 1.196)	0.94 (0.79, 1.12)	0.97 (0.81, 1.15)	1.05 (0.88, 1.26)

^1^ IRR = incidence rate ratio; CI = confidence interval; * *p* < 0.001; ** *p* < 0.05.

## Data Availability

Not applicable.

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
