# Peer review of "The Pandemic-Related Factors Associated with Emergency Department Visits in Portugal throughout Two Years of the Pandemic: A Retrospective Population-Based Study"

_ijerph, 2023, doi:10.3390/ijerph20021207_

Round 1

Reviewer 1 Report

This study sought to examine the COVID-19 impacts on emergency department visits in Portugal. The Intro covers relevant literature from Europe about pandemic impacts on ED visits. Portugal-focused studies are mentioned here (refs 9 and 10) and it may be worth providing a little more info on these (method, population etc.) so that the uniqueness of the current study can be further highlighted. Can you please provide an explanation of what is meant by 'mobility and visits' in this context? Is it simply people not moving around and visits to other people in general? Or is it more specific to visits to healthcare services?

The dataset used seems to provide a comprehensive overview of ED use within Portugal. As someone not based in Portugal, does data capture of mainland Portugal cover most of the country? Or are significant parts of Portugal not captured? Can a little more info be provided on this please. I am not 100% sure about the 'Google search' variable indicating levels of interest in the topic, but understand why it was used. I am curious about how the score of 0 to 100 was obtained for this measure though - is that something that Google creates or did the authors create this? Some further info about how that score is generated would be nice. 

Use of maximum values during the month as the values used for analysis seems odd. Why was the average or median score not used instead? Can this decision be justified please? Statistical methods seem correct based on the aims of the study and the data being used. Line 167-169 - probably don't need to present the numbers in brackets as they are clear in the Table. 

I would avoid re-presenting statistics in the Discussion - keep these to the Results section only. Lines 253-255 - is that specifically in Portugal or among studies of any country? A study limitation is noted as small sample size. However I don't actually see a sample size ever noted in the paper - can this information be included in the paper please? Why was the sample size small given this is a national study or administratively collected data? The limitations based on the Google variables are a little alarming also, given they are used as key variables in this analysis. Can you perhaps provide some information about how much of the population of Portugal uses Google as its main website to collect information (if such info exists)? 

Line 43 = 'EDs' not Eds'. 

Reviewer 2 Report

Review of manuscript no ijerph-2049964 entitled “The Pandemic-Related Factors Associated with Emergency Department Visits in Portugal through Two Years of the Pandemic: A Retrospective Population-Based Study

The aim of the reviewed manuscript was to investigate the pandemic-related factors associated with the number of emergency departments (ED) visits, in mainland Portugal and each of its regions. Authors collected data on ED visits from the Portuguese National Health Service Transparency Portal. Data on daily cases of COVID-19 and weekly vaccination for the mainland and the regions were extracted from the GitHub repository. Authors analyzed data from March 2020 to March 2022. Also, data on mobility, containment index and Google search volume were retrieved from open online sources at different time points. The quasi-Poisson generalized linear regression model was used in statistical analysis of data.  

All part of manuscript were described and presented correctly. The results are not revealing or surprising, but they are the Authors' contribution to the analysis of the COVID-19 pandemic in the Portuguese population. The article may be considered  for publication in the IJERPH  journal.

Reviewer 3 Report

This paper discussed an interested topic with reliable data. The conclusions obtained are useful. Suggested author to discuss the impact of environment factor.
